# Gastrointestinal Involvement in Extra-Digestive Disease: Which Is the Role of Fecal Calprotectin?

**DOI:** 10.3390/medicina58101384

**Published:** 2022-10-02

**Authors:** Angela Saviano, Marcello Candelli, Christian Zanza, Andrea Piccioni, Alessio Migneco, Veronica Ojetti

**Affiliations:** 1Department of Emergency Medicine, Fondazione Policlinico Universitario A. Gemelli, IRCCS, 00168 Rome, Italy; 2Department of Internal Medicine, Ospedale San Carlo di Nancy, 00165 Rome, Italy; 3Department of Emergency Medicine, Policlinico Gemelli/IRCCS University of Catholic of Sacred Heart, 00168 Rome, Italy

**Keywords:** fecal calprotectin, psoriasis, behcet’s disease, atopic dermatitis, parkinson’s disease, alzheimer’s disease, COVID-19

## Abstract

Fecal calprotectin (FC) is a very sensitive marker of inflammation of the gastrointestinal tract. Its clinical utility can be appreciated in both intestinal and extraintestinal diseases. Recent evidence suggests a link between intestinal inflammation and dermatological, rheumatic and neurological diseases. This review focuses on the role of FC in non-gastrointestinal disease, such as rheumatic, dermatologic, neurologic and last but not least SARS-CoV-2 infection.

## 1. Introduction

Calprotectin (CP) is a dimer composed of S100A8 and S100A9, which are calcium and zinc binding proteins. CP is found mainly in neutrophils, where under constitutive conditions, it represents about 45% of the total cytosolic protein. Moreover, calprotectin is constitutively expressed by monocytes, macrophages, dendritic cells oral keratocytes and squamous mucosal epithelium. In inflammation, the expression of calprotectin is increased. CP is released by neutrophils, monocytes, and macrophages [1] during inflammation due to its antimicrobial properties. CP can be detected in serum, urine, cerebrospinal, synovial, and pleural fluids in proportion to the degree of any existing inflammation, but the most useful and widely used form is in stool as a reliable marker of intestinal tissue inflammation [2]. Moreover, CP concentration in feces is approximately six times higher than in plasma. Elevated levels of fecal calprotectin (FC) can be detected in both primary gastrointestinal disorders and extraintestinal diseases [3]. FC can be easily measured by enzyme-linked immunosorbent assay (ELISA) [4]. FC is elevated in over 95% of patients with inflammatory bowel disease (IBD) and correlates with clinical disease-activity [5]. FC has high specificity and sensitivity in predicting clinical relapse [5]. Elevated levels of FC have also been found in patients with NSAID-induced enteropathy [6] and used for the diagnosis of colorectal cancer [7]. It is also a useful test to differentiate between abdominal pain secondary to IBD and irritable bowel syndrome (IBS) [4,5]. Recent evidence suggests that intestinal inflammation correlates with dermatologic, neurologic, and rheumatologic diseases [8,9]. In fact, the intestinal mucosa and the skin are barriers that share many similarities. Both the skin and the gut are in contact with the environment and are the first line of defense against external agents and pathogens. Both are rich in immune cells and are colonized by microbial cells [8]. Both are niches for fungi, viruses and especially bacteria, which are essential for metabolic and immune functions. The gut–skin axis can be considered an integral part of the gut–brain–skin axis [9,10]. In addition, research on the role of gut microbiota and its correlation with brain function and neurological disorders has increased. The imbalanced composition of gut microbiota observed in these diseases, with an overexpression of proinflammatory bacteria determine an increase of leaky gut that may contribute to intestinal inflammation [11,12]. Similarly, the human microbiome is a key factor in the development of autoimmune and rheumatologic diseases due to its involvement in immune homeostasis [11,13]. This review focuses on the important role of FC in many extra gastrointestinal diseases from rheumatologic to dermatologic to neurologic ones to the recent SARS CoV-2 infection Figure 1.

## 2. Fecal Calprotectin and Rheumatologic Diseases

### 2.1. Ankylosing Spondylitis

Ankylosing spondylitis (AS) is a chronic rheumatic disease characterized by inflammation and ankylosis of the joints (e.g., sacroiliac joints), spine, peripheral joints and entheses but it also affects the eyes, urinary tract, gut and heart. There is a close relationship between spondyloarthropathy (SpA) and IBD. In both diseases the immune system dysregulation, the gut dysbiosis, and genetic factors (HLA B27) play an important role in the development and pathogenesis [11]. Moreover, IBD may be associated with sacroiliitis, arthritis and uveitis. Approximately 50% of patients with SpA have microscopic bowel lesions on colonoscopy and this gut inflammation is associated with more severe axial disease. An interesting hypothesis is that the presence of bowel inflammation with HLS B12 positivity allows the contact between bacterial antigen and the immune system leading to a possible cross mimicry reaction between joints, bone, cartilage and bacteria that contribute to AS-development. Many studies have found an association between FC and SpA. One of the first studies showed that in more than 200 patients, 70% of enrolled subjects with AS had elevated levels of FC greater than 50 mg/kg and 30% showed levels higher than 200 mg/kg. Moreover, the levels of FC were significantly related to the number of tender joints affected, increasing age of patients, and disease duration. On the other hand, all patients had low or normal levels of serum calprotectin [14]. Kang et al. [15] performed a study on 190 patients with axSpA and found that the levels of FC were related to the activity of the disease. Specifically, high FC levels were more closely related to peripheral joint inflammation than to axial joint inflammation [15]. Furthermore, Gazim et al. [16] measured the levels of FC in a series of patients with anterior uveitis and AS, AS alone, and uveitis of other etiologies to determine whether anterior uveitis with AS had higher levels compared with other groups. In this cross-sectional study performed on 28 patients subjects with AS and both AS and uveitis were found to have elevated levels of FC compared with patients with uveitis, leading them to conclude that the dosage of FC may be useful in distinguishing uveitis associated with spondylarthritis from uveitis of other etiologies. This confirms the close link between AS and colitis. The Spartacus study (2019) [17] was a cross-sectional study that compare patients with axial ankylosing spondylitis (axSpA) and with AS to a control group. Patients with AS had higher FC values than subjects with non-axSpA, and both higher than control group. The authors of this study also determined anti-Saccharomyces cerevisiae antibodies (ASCA; IgA and/or IgG types) another biomarker used in the diagnosis of IBD, in particular in Crohn’s disease [17]. Both IgG and IgA ASCA antibody tended to be normal in the three groups with no significant differences among groups. This study proved that elevated FC was associated with greater disease activity and deterioration of physical functions and may be a marker of more severe disease. Another interesting paper examined the prevalence of intestinal inflammation using FC levels in patients with AS compared with that in patients with rheumatoid arthritis (RA) and in patients with non-inflammatory rheumatic diseases. A total of 194 patients participated in this study. The AS group had significantly higher values of FC than the RA group, although the levels of FC in this study did not correlate with disease activity in patients with AS [16,17,18]. A systematic review and meta-analysis published in 2020 on the role of calprotectin in AS concluded that both serum and stool levels are elevated in patients affected by AS [19]. A very recent paper by Emad et al. [20] evaluated the clinical utility of FC in patients with differentiated and undifferentiated AS. The study included a total of 52 patients with differentiated SpA and 33 patients with undifferentiated SpA who were compared with 50 matched controls. The mean value of FC was significantly higher in the differentiated SpA patients compared with the undifferentiated and control groups (*p* < 0.001). These data confirm the role of FC as a biomarker for gut inflammation in patients with AS [20].

### 2.2. Behcet Diseases

Behçet’s disease (BD) is a systemic inflammatory vasculitis of unknown etiology. It is characterized by recurrent oral and genital ulcers, skin lesions, and ocular inflammation. In some patients has been observed the involvement of the vascular, neurologic, and gastrointestinal (GI) systems. In their work, Özşeker et al. [21] estimated the usefulness of FC in the detecting intestinal involvement in patients with BD. Their study included 30 subjects affected by BD and 25 healthy volunteers as a control group. FC was the only statistically significantly increased marker of inflammation in patients with BD compared to the control group. No statistically significant differences were found in other inflammatory markers such as C reactive protein (CRP). Subjects with intestinal involvement as ileitis and ulcers in the terminal ileum showed a significantly increase level of FC compared to the group with negative intestinal involvement [21]. Another study confirms these results and suggest FC as a useful tool for diagnosis gastrointestinal involvement in patients with BD. Fecal and serum calprotectin, CRP levels, and colonoscopy were performed in 39 patients with BD. FC, but not serum calprotectin seemed to be a useful noninvasive tool to evaluate disease activity. Moreover, in a multivariate analysis, the FC test was the only significant predictor of remission in patients with [22].

## 3. Fecal Calprotectin and Dermatological Diseases

### 3.1. Psoriasis

Psoriasis is a common, chronic, and inflammatory skin disease with a strong genetic predisposition and autoimmune pathogenesis. Skin lesions occur on the knees, elbows, trunk and scalp and are characterized by red, itchy, and scaly patches [23,24]. Inflammation is not limited to psoriatic skin but may involve various organ systems. Therefore, it has been postulated that psoriasis is a systemic entity rather than a simple dermatologic disease [25]. Patients with psoriasis have a leaky gut, which causes increased permeability with increased blood concentrations of metabolites derived from the gut microbiota. This is responsible for the systemic inflammation. The new therapeutic approach based on these mechanisms, attempts to modulate the intestinal barrier and reduce gut inflammation [26]. Few studies investigated the level of FC in patients with psoriasis. A very old paper studied the effects of short-term oral treatment with seal oil in 40 patients with psoriatic arthritis (PsA) due to its anti-inflammatory property. It is known that many patients with PsA have signs of intestinal inflammation, which may play a role in development of the disease [27]. They determine the levels of FC in 40 patients with PsA, finding that 9 (21%) had elevated values, suggesting bowel inflammation or asymptomatic colitis. In addition, compared to CRP, FC was the only markers strongly correlated with the number of tender joints involved by the disease (*p* < 0.05) [27]. Recently, a small pilot study of 10 patients on the effects of probiotic strains on disease activity and enteric permeability in psoriatic arthritis was published. They evaluate FC as a marker of intestinal inflammation and fecal zonulin, α1-antitrypsin as markers of gut permeability. They found high levels of FC in 6 (60%) patients and high levels of fecal zonulin in 10 (100%), while abnormally high levels of α1-antitrypsin were found in 6 (60%) patients. Interestingly, a significant decrease of levels of fecal zonulin was observed 4 weeks after probiotic therapy, while no statistically significant difference was found in α1-titrypsin levels, albeit with a decreasing trend after therapy, suggesting a beneficial effect of probiotic intake on intestinal permeability. FC decreases in over 30% of patients suggesting also a reduction in gut inflammation. Both this effect may affect the disease activity [28].

### 3.2. Atopic Dermatitis

Atopic dermatitis (AD) is a chronic inflammatory disorder that affects over 20% of children. Several factors are involved in the development of AD such as an imbalance of the gut microbiota, the dysfunction of the skin barrier, the dysregulation of the immune system, and other environmental factors. Dysbiosis and the resulting intestinal inflammation seem to play a crucial role in the development of allergic diseases. For this reason, in the recent years, several authors have attempted to evaluate the role of FC as a marker of intestinal inflammation in children affected by AD. Seo et al. [27] studied 65 children with AD and a control group. In 32.3% of enrolled subjects FC was higher than 50 μg/g but they found no significant difference in age, sex, body mass index (BMI), and birth weight between the high or low FC groups. Interestingly, children with higher values of FC showed a more severe AD, a higher blood eosinophil and IgE levels [27]. Orivuori et al. [28] demonstrated in their study that children with a high value of FC at 2 months of age had an increased risk of developing AD later. This risk could be due to the imbalance of gut microbiota in the early childhood determining an increased intestinal inflammation which affects the immune system and promote to the development of AD.

## 4. Fecal Calprotectin and Neurological Diseases

### 4.1. Parkinson’s Disease

Parkinson’s disease (PD) and multiple system atrophy (MSA) are neurodegenerative disorders in which there is an accumulation of insoluble α-synuclein protein in the nervous system, in neurons (PD), or in the glial cells (MSA). Constipation is a very common gastrointestinal (GI) symptom in these patients, and the pathological α-synuclein has been found in the enteric nervous system in both diseases [29,30]. The so-called “gut-brain axis” seems to be involved in the pathogenesis of these disorders, especially in PD. A significant intestinal inflammation associated with increased expression of proinflammatory cytokines has been found in colonic biopsies from PD patients [30]. A causal relationship between the gut and PD has not been established but intestinal dysbiosis, with gut imbalance is common in PD. Moreover, the Resista-PD Trial [31] showed that in PD-patients with a prevalence of proinflammatory bacteria, elevated levels of FC were detected. In addition, epidemiological data suggest an association between IBD and PD. Hor et al. [32] published an interesting paper highlighting the increased levels of FC in patients with PD and MSA compared to controls. The levels are higher in MSA and in patients older than 65 years. These data support the presence of intestinal inflammation in these neurological disorders. At the same time, they found no correlations between FC and gender [32] or PD duration [32]. In a case-control study, Schwiertz et al. [29] examined the levels of FC and lactoferrin as markers of intestinal inflammation and alpha-1-antitrypsin and zonulin as markers of intestinal permeability in patients with PD and in healthy controls. All of these markers were significantly elevated in PD, but none of the four fecal markers correlated with disease severity [29]. Sturgeon et al. [33] performed an interesting study in which they determined the values of FC as a biomarker of intestinal inflammation and the value of zonulin as a biomarker of intestinal barrier disruption in PD patients. They also evaluated if Fc and zonulin can be useful tools in the diagnosing and for management of PD as is common performed in IBD. The main value of FC was significantly higher in PD patients compared to the controls: 54.5 vs. 9.7 ng/mL (*p* < 0.0001). Zonulin fecal value (ng/mL) was also higher in PD patients than in controls, but without reaching a statistical significance [4]. Dumitrescu et al. [34] evaluated both serum and fecal markers of intestinal inflammation and intestinal barrier permeability in 22 patients with sporadic PD and 16 healthy patients. They observed significantly high serum and fecal calprotectin in patients affected by PD. They also found increased serum and stool zonulin levels in PD [34].

### 4.2. Alzheimer’s Disease

Alzheimer’s disease is a common neurodegenerative disorder, characterized by the accumulation of extracellular aggregates of amyloid-β (Aβ) plaques in the cortical and limbic brain areas of the human brain. The intestinal inflammation seen in Alzheimer’s disease affected patients leads to an increase in calprotectin, which may contribute to the formation of amyloid fibril in both the gut and the central nervous system. Calprotectin is composed by two distinct subunits S100A8 and S100A9, which are capable of forming amyloid oligomers and fibrils very similar to α-syn and Aβ. Leblhuber et al. [35] determined the level of FC in a group of 22 patients with Alzheimer’s disease and found that it correlated with the level of aromatic amino acids value. They observed that 73% of the patients had high level of FC and the concentrations correlated inversely with the serum levels of tryptophan, tyrosine, and phenylalanine (*p* < 0.05) [35]. Moreover, Horvath et al. [36] discovered a high level of S100A9 subunit in cerebrospinal fluid of patients with Alzheimer’s disease. On the other hand, Stolzenberg et al. [37] discovered a high expression of α-syn in the inflamed intestinal mucosa with a strong attraction to leukocytes (neutrophils and monocytes), which determine the immune response. A possible role of leaky gut in the pathogenesis of Alzheimer’s disease was postulated by Kohler et al. [38,39]. The presence of dysbiosis may in some subjects determined a leaky gut with a translocation of bacteria, thus increasing inflammation and accumulation of Aβ. Recent study has also hypotized an implication of *H. pylori* infection in determining an alteration of gastric pH, thus influencing gut microbiota composition and promoting dysbiosis with an increase of proinflammatory bacteria such as *Proteobacteria* and *Enterobacteria* [39].

## 5. Fecal calprotectin and COVID-19

FC produced by neutrophils and studied in detail in inflammatory bowel diseases, has been used to evaluate the inflammatory process in the gut during COVID-19 related pandemic. One of the first observational study performed by our group on 65 COVID-19 positive patients in the emergency department showed that 30% had high level of FC with a mean of 70 µg/g. Of these patients, 60% had pneumonia compared with 10% in the group with normal FC. Patients with normal FC were male and younger than patients with high calprotectin level. In our study we found that elevated FC was independently associated with a pathologic chest radiograph (OR 11.2 [CI 95% 1.29–28.2], *p* 0.028). Conversely, age, sex, and gastrointestinal symptoms were not independently associated with an increased value of FC. Thus, we conclude that the digestive system could be a potential pathway for SARS-CoV-2 infection and that monitoring FC levels could help clinicians identify the degree of inflammation during infection and potential COVID-19 progression. COVID-19 has a variable, so more specific diagnostic tools are needed to detect severe cases and predict the outcome of the infection. FC could be a new tool to stratify patients in terms of severity [40]. Moreover, new studies are needed to explore this area and find a marker that is easy and simple to collect and useful for following-up patients who had COVID-19. Another work that investigated the relationship between serum levels and FC in patients with COVID-19 found no differences between these two markers and the presence of GI symptoms but showed a positive correlation of SC and FC with disease diagnosis and prognosis severity [41]. A recent systematic review and meta-analysis analyzed ten studies (eight quantitative and two qualitative) that shoved that circulating calprotectin levels were elevated in patients with COVID-19 [42,43]. More importantly, these higher calprotectin levels could distinguish severe COVID-19 form from non-severe forms, and thus predict poor outcome as well as predict gut inflammation in patients with COVID-19. Therefore, calprotectin can be considered a useful biomarker for COVID-19 that has both diagnostic and prognostic significance. However, further research in this field is needed.

## 6. Conclusions

The clinical utility of FC can be showed in both intestinal and non-intestinal diseases, highlighting the role and the connection of the human gut with other body systems (brain, skin, lungs, joints, etc.) Figure 2. FC is an easy marker, quick to perform that can be repeated and without risk. The determination of FC could be performed also in patients who primarily do not show gastrointestinal symptoms; in fact, it has a good correlation with activity and duration of some dermatological, neurological and rheumatological diseases. Gut, in fact, can be considered as a “mirror” of other diseases. This review aims to open the way for neurologists, rheumatologists, dermatologists and specialist of infectious diseases to request this test for a better management and clinical monitoring of patients affected by the diseases described above, even if they don’t primary show gastrointestinal manifestations. We believe this review could be a starting point for conducting new research, useful also in the daily clinical practice.

## Figures and Tables

**Figure 1 medicina-58-01384-f001:**
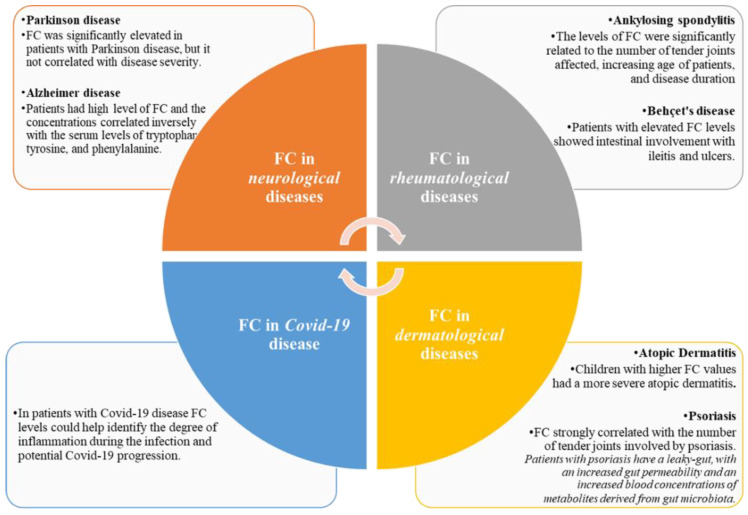
Fecal calprotectin in extra gastro-intestinal diseases.

**Figure 2 medicina-58-01384-f002:**
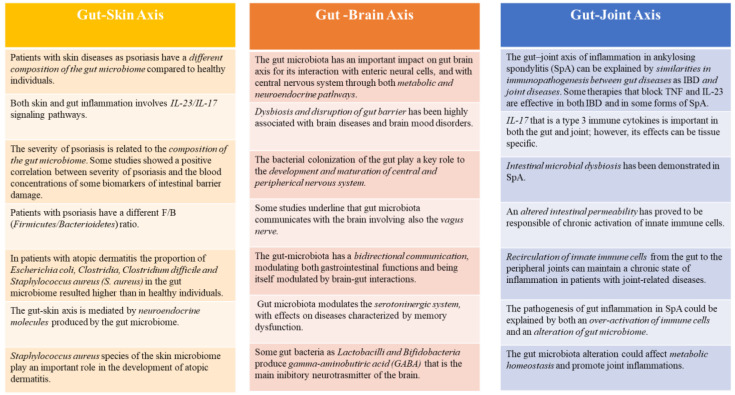
Gut-Skin, Gut-Brain, Gut-Joint Axis and the role of fecal calprotectin. *Faecal calprotectin*, as a marker of gut inflammation, can be very useful for clinicians in both intestinal and non-intestinal diseases, due to the link between gut and other systems as skin, brain, joints.

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
