# Peer review of "Gastrointestinal Involvement in Extra-Digestive Disease: Which Is the Role of Fecal Calprotectin?"

_medicina, 2022, doi:10.3390/medicina58101384_

Round 1
Reviewer 1 Report
1. The authors did not answer the question that was formulated in the title – what is the role of fecal calprotectin (FC). The article provides data on its increase in a number of nosologies and its correlation with various markers of permeability.
2. The authors indicate that an increase in the level of fecal calprotectin (line 24, 25) is detected both in diseases of the gastrointestinal tract and in extraintestinal pathology. Further (line 38-40), the authors write that «research on the role of gut microbiota, brain function and neurological 39 disorders has increased, revealing that an imbalanced composition of gut microbiota may contribute to some neurological disorders». At the same time, it is known that fecal calprotectin is primarily a marker of inflammation, and not changes in the composition of the intestinal microbiota. Thus, the reader has a misconception that changes in intestinal microbiocenosis leads to inflammation and an increase in the level of fecal calprotectin.
3. Lines 28-35: there are no references to literary sources.
4. The article gives the impression of incompleteness: describes an increase in FC in ankylosing spondylitis, Behcet's disease, psoriasis, atopic dermatitis, Parkinson's disease, Alzheimer's disease and Covid-19 disease; does not give an algorithm of action for a practitioner in using the data obtained; does not allow verification of the inflammatory process; does not give systemic knowledge about the role of intestinal permeability and changes in its microbiota under various nosologies.
5. It is necessary to supplement the article with a section on the microbiota, its changes, about the mucosal barrier. Add a visual series in the form of a figure (diagrams, pictures).
6. Expand the final part with an emphasis on practical activities and significance.
Author Response
- The authors did not answer the question that was formulated in the title – what is the role of fecal calprotectin (FC). The article provides data on its increase in a number of nosologies and its correlation with various markers of permeability.
As suggested We modified the title
- The authors indicate that an increase in the level of fecal calprotectin (line 24, 25) is detected both in diseases of the gastrointestinal tract and in extraintestinal pathology. Further (line 38-40), the authors write that «research on the role of gut microbiota, brain function and neurological 39 disorders have increased, revealing that an imbalanced composition of gut microbiota may contribute to some neurological disorders». At the same time, it is known that fecal calprotectin is primarily a marker of inflammation, and not changes in the composition of the intestinal microbiota. Thus, the reader has a misconception that changes in intestinal microbiocenosis leads to inflammation and an increase in the level of fecal calprotectin.
As suggested, we better specify the concepts
- Lines 28-35: there are no references to literary sources.
We added references
- The article gives the impression of incompleteness: describes an increase in FC in ankylosing spondylitis, Behcet's disease, psoriasis, atopic dermatitis, Parkinson's disease, Alzheimer's disease and Covid-19 disease; does not give an algorithm of action for a practitioner in using the data obtained; does not allow verification of the inflammatory process; does not give systemic knowledge about the role of intestinal permeability and changes in its microbiota under various nosologies.
Unfortunately, at this moment, there isn’t an algorithm including the determination of faecal calprotectin in patients affected by non-gastrointestinal disease but this review could be a starting point for other specialists such as rheumatologist, dermatologist, neurologist etc. to be familiar with gastrointestinal exams.
We thank you for your suggestion but we believe that the role of intestinal permeability and changes in its microbiota (under various diseases) could be the topic of another review.
- It is necessary to supplement the article with a section on the microbiota, its changes, about the mucosal barrier. Add a visual series in the form of a figure (diagrams, pictures).
We added two figures
- Expand the final part with an emphasis on practical activities and significance.
We expanded the final part as suggested
Reviewer 2 Report
The manuscript presented for evaluation is a review. Meets the requirements for this type of publication. The subject of the work is topical. The authors focused on selected clinical aspects related to the assessment of calprotectin concentration. In my opinion, the manuscript is interesting, informative and worth publishing.
Author Response
Thank you for your comment
Reviewer 3 Report
In this manuscript, Angela et al. describe a review of faecal calprotectin in extra gastro-intestinal diseases. This review introduced the role of FC in extra-gastrointestinal diseases from rheumatological, dermatological, neurological ones and SARS-CoV-2 infection. Angela et al. believed that the clinical utility of faecal calprotectin can be showed in both intestinal and nonintestinal diseases, highlighting the role and connection of the human gut with other body systems (brain, skin, lungs, joints, etc.) and opening new research scenarios. The overall subject is meaningful and worthy of study. I feel that it is suitable for publication in this journal but, after the authors should accept few revisions of their paper, particularly on the following points:
This review lacks the negative results of FC in these diseases, and it is appropriate to introduce the non significant related studies of FC in these diseases, because this will enable readers to have a more comprehensive understanding of the role of FC in these diseases.
L29-30, page1: A reference should be added after this sentence.
L70, page2: The format of this reference “(13)”should be consistent with that of the entire manuscript.
L144, page3; L192, page4: The abbreviations of Atopic dermatitis(AD) and Alzheimer’s disease (AD) were the same in this manuscript. In order to avoid confusion, they should be distinguished. For example, one is not abbreviated.
L207, page5: There is no full stop after this sentence.
L209-212, page5: It seems that the results of H. Pylori's study are not closely related to the theme of the manuscript, that is, the role of FC.
Author Response
In this manuscript, Angela et al. describe a review of faecal calprotectin in extra gastro-intestinal diseases. This review introduced the role of FC in extra-gastrointestinal diseases from rheumatological, dermatological, neurological ones and SARS-CoV-2 infection. Angela et al. believed that the clinical utility of faecal calprotectin can be showed in both intestinal and nonintestinal diseases, highlighting the role and connection of the human gut with other body systems (brain, skin, lungs, joints, etc.) and opening new research scenarios. The overall subject is meaningful and worthy of study. I feel that it is suitable for publication in this journal but, after the authors should accept few revisions of their paper, particularly on the following points:
This review lacks the negative results of FC in these diseases, and it is appropriate to introduce the non- significant related studies of FC in these diseases, because this will enable readers to have a more comprehensive understanding of the role of FC in these diseases.
As suggested we added some studies.
L 29-30, page 1: A reference should be added after this sentence.
We added references
L70, page 2: The format of this reference “(13)”should be consistent with that of the entire manuscript.
We modified as suggested
L144, page3; L192, page4: The abbreviations of Atopic dermatitis(AD) and Alzheimer’s disease (AD) were the same in this manuscript. In order to avoid confusion, they should be distinguished. For example, one is not abbreviated.
We modified
L207, page5: There is no full stop after this sentence.
We added
L209-212, page5: It seems that the results of H. Pylori's study are not closely related to the theme of the manuscript, that is, the role of FC.
We modified
Round 2
Reviewer 1 Report
The article has been well finalized, figures and literary references have been added, explanations have been made. But I still have one question for the authors: what do the arrows in the center of figure 1 mean?
Author Response
But I still have one question for the authors: what do the arrows in the center of figure 1 mean?
We wanted to express that fecal calprotectin may play a role in dermatological, rheumatological, COVID-19, neurological diseases... but if they create confusion we can remove them
Reviewer 3 Report
I think it is suitable for publication in this journal after the author's modification.
Author Response
I think it is suitable for publication in this journal after the author's modification.
Thank you,
during the first submission there was a mistake in saving name of authors but then we sent the "Change of Authorship Form" signed as requested